

# First three new species of free-living marine nematodes of the *Molgolaimus* (Nematoda: Desmodoridae) from the continental shelf of the Brazilian coast (Atlantic Ocean)

Alex Manoel[1,2], Patrícia F. Neres[1] and Andre M. Esteves[1,2]

[1] Zoologia, Universidade Federal de Pernambuco, Recife, PE, Brazil
[2] Center of Biosciences, Universidade Federal de Pernambuco, Recife, PE, Brazil

## ABSTRACT

Three new species of the *Molgolaimus* (Nematoda: Desmodoridae) are described from sample sediments collected in the South Atlantic, along the continental shelf break of Northeastern Brazil. This is the first time that new species of *Molgolaimus* have been described from sample sediments collected in the Brazilian coast. *Molgolaimus sigmoides* **sp. nov.** is characterized by four small cephalic sensillae, a buccal cavity with three small teeth, S-shaped spicules and gubernaculum with dorsal-caudal apophysis. *Molgolaimus paralongispiculum* **sp. nov.** possesses four setiform cephalic sensillae, a buccal cavity with three teeth, thin and elongated spicules and gubernaculum with anteriorly oriented apophysis. *Molgolaimus brevispiculum* **sp. nov.** is characterized by its possession of four setiform cephalic sensillae, an unarmed buccal cavity, short spicules and absent gubernaculum. We propose to amend the diagnosis of the genus, redistribute *Molgolaimus* species into subgroups $1b_1$ and $1b_2$, and to rearrange the order of presentation of subgroups 4a and 4b.

# INTRODUCTION

The genus *Molgolaimus Ditlevsen, 1921* was erected based on two specimens obtained from sample sediments collected in the Auckland Islands, New Zealand, by Dr. Th. Mortensen during his expedition to the Pacific between 1914 and 1916 (*Ditlevsen, 1921*). *Molgolaimus* was originally classified within Microlaimidae (*Micoletzky, 1922*; *Gerlach & Riemann, 1973*; *Gerlach & Riemann, 1974*) presumably based on similarities to the genus *Microlaimus De Man, 1880* in head and amphidial fovea shape, arrangement of head sensilla, and buccal cavity structure (*Leduc, Fu & Zhao, 2019*). Since then, the taxonomic position of the genus has been investigated and modified over time, both based on morphology and phylogenetic analyses of the small subunit (SSU) sequences.

*Jensen (1978)* reviewed Microlaimidae and, based on a series of morphological characteristics, including the morphology of the gonads of males and females, erected the family Molgolaimidae *Jensen, 1978* to accommodate two subfamilies: Aponematinae

Corresponding author
Andre M. Esteves,
andresteves.ufpe@gmail.com

*Jensen, 1978*, with a single genus *Aponema Jensen, 1978* and Molgolaiminae *Jensen, 1978*, for *Molgolaimus* and *Prodesmodora Micoletzky, 1923* (*Jensen, 1978*; *Muthumbi & Vincx, 1996*). The other genera existing until then (*Bolbolaimus Cobb, 1920*; *Calomicrolaimus Lorenzen, 1976*; *Ixonema Lorenzen, 1971a* and *Microlaimus*) remained within Microlaimidae.

In his phylogenetic analyses, *Lorenzen (1981)*, based on the morphology of the gonads of males and females, disagreed with *Jensen*'s (*1978*) establishment of the family Molgolaimidae and downgraded the family as a subfamily Molgolaiminae in Desmodoridae *Filipjev, 1922* (*Muthumbi & Vincx, 1996*; *Shi & Xu, 2016*). The same author reintegrated *Aponema* into Microlaimidae and moved *Prodesmodora* to the subfamily Prodesmodorinae (*Lorenzen, 1981*). In the same study, Lorenzen placed *Molgolaimus* in its own single-genus subfamily within the Desmodoridae and established the monophyly of the superfamily Desmodoroidea (*Filipjev, 1922*) (Order Desmodorida *De Coninck, 1965*) based on the presence of only one anterior testis in males (*Leduc, Fu & Zhao, 2019*).

*Leduc, Verdon & Zhao (2018)*, based on SSU sequences phylogenetic analyses which placed *Molgolaimus demani Jensen, 1978* in a monophyletic clade with Microlaimidae sequences, proposed that *Molgolaimus* be removed from Desmodoroidea and placed within Microlaimoidea *Micoletzky, 1922*. The same authors also proposed that the family Molgolaimidae be reinstated and moved to accommodate *Molgolaimus* within Microlaimoidea. Additionally, they erected the order Microlaimida *Leduc, Verdon & Zhao, 2018* to accommodate the superfamily Microlaimoidea. The order Microlaimida, based on the results obtained by *Leduc, Verdon & Zhao, 2018*, began to include the families Microlaimidae, Molgolaimidae, Monoposthiidae *Filipjev, 1934* and Aponchiidae *Gerlach, 1963*.

However, *Leduc, Fu & Zhao (2019)* explained that the molecular analysis provided by *Leduc, Verdon & Zhao (2018)* ignored the fact that *Molgolaimus demani* was synonymized with *Microlaimus tenuispiculum De Man, 1922* by *Lorenzen (1981)*, *Lorenzen (1994)* based on the structure of the reproductive system. *Leduc, Fu & Zhao (2019)*, using SSU sequences from *Molgolaimus kaikouraensis Leduc, Fu & Zhao, 2019* and another species, referred to as *Molgolaimus* sp., performed new molecular analyses and found no support for placing *Molgolaimus* with either the Desmodorida or Microlaimida. The results of SSU phylogenetic analysis obtained by *Leduc, Fu & Zhao (2019)* suggest that *Molgolaimus* should be classified with Chromadorida *Chitwood, 1933*. Recently, *Sun & Huang (2024)* carried out molecular analyses using SSU sequences from *Molgolaimus longicaudatus Sun & Huang, 2024*. The results obtained by these authors showed that *Molgolaimus longicaudatus* clustered within Desmodoridae clade, supporting the opinion of Lorenzen.

In order to facilitate the identification of the different species of the genus, *Fonseca, Vanreusel & Decraemer (2006)* distinguished four groups of *Molgolaimus* species based on a frequency distribution for spicules length (Group 1: species with spicules <35 µm; Group 2: spicules ranging between 35 and 53 µm; Group 3: spicules ranging between 53 and 80 µm; Group 4: spicules longer than 80 µm). Groups 1 and 4 were divided into subgroups (1a, $1b_1$, $1b_2$ and 1c; 4a and 4b, respectively) based on relative spicules length, as well as body length and ratios of body dimensions.

**Table 1** **Collection stations, their respective coordinates and depth.** The samples were collected at the break of the continental shelf in Northeast Brazil, South Atlantic.

| Station | Latitude | Longitude | Depth |
|---------|----------|-----------|-------|
| 2 | S05°42′54.42″ | W34°59′31.92″ | 60 m |
| 4 | S06°27′06.06″ | W34°45′53.64″ | 56 m |
| 9 | S08°38′20.46″ | W34°45′45.12″ | 47 m |
| 10 | S08°56′36.78″ | W34°50″16.02″ | 54 m |
| 11 | S09°15′30.54″ | W34°57′13.14″ | 87 m |
| 16 | S10°44′59.28″ | W36°25′32.88″ | 58 m |
| 17 | S11°00′00.54″ | W36°49′58.98″ | 54 m |
| 23 | S13°04′10.32″ | W38°25′46.98″ | 65 m |

*Molgolaimus* species are present in all oceans, ranging from shallow water regions to the deep sea (*Fonseca, Vanreusel & Decraemer, 2006*), including records in estuarine areas (*Warwick, 1970*; *Zhou et al., 2020*; *Sun & Huang, 2024*). Along the Brazilian coast, at a generic level, this taxon was previously recorded in articles published in indexed journals and dissertations/thesis for different habitats: continental shelf adjacent to the Santos Estuarine System, São Paulo (*Yaginuma, 2011*); Slope (*Moura, 2013*) and Canyons and adjacent areas (*Silva, 2012*) in the Campos Basin, Rio de Janeiro and in the Capibaribe River estuary, Pernambuco (*Cavalcante, 2023*). At a specific level, recorded the presence of *Molgolaimus abyssorum Muthumbi & Vincx, 1996* and *Molgolaimus lazonus Vitiello, 1970*; *Jensen, 1978* in a deep-sea region of the Campos Basin, Rio de Janeiro (*Lira, 2005*; *Venekey, 2017*).

In the present study, representatives of the genus *Molgolaimus* were found from samples collected in the South Atlantic, along the break of the Continental Shelf in Northeast Brazil. Here we describe three new species of *Molgolaimus*. This is the first time that new species of this taxon have been described from sample sediments collected on the Brazilian coast. We also propose to amend the diagnosis of the genus, redistribute *Molgolaimus* species into subgroups $1b_1$ and $1b_2$, and to rearrange the order of presentation of subgroups 4a and 4b.

## MATERIAL AND METHODS

**Study area and sampling.** This information were previously described in *Manoel, Neres & Esteves (2024)*. Table 1 presents details of the collection stations relevant to this study. For sediment collection, a box-corer was used, while meiofauna samples were collected with a corer with dimensions of 10 cm × 10 cm. The sediment samples were taken in triplicate. The material collected was placed in plastic pots and fixed with 4% formaldehyde.

**Sample processing.** In laboratory, sediment samples were washed with filtered water through sieves of 0.5 mm and 0.045 mm mesh. The remaining samples in the smaller

mesh sieve were extracted with SICOL-40 colloidal silica solution (specific gravity 1.18) (*Somerfield, Warwick & Moens, 2005*).

Under a stereomicroscope, Nematodes was sorted out and deposited in a small glass container containing Solution 1 described by *De Grisse (1969)*. The specimens were then transferred to glycerin and diaphanized using the methodology described by *De Grisse (1969)*, and subsequently mounted permanently on glass slides (*Cobb, 1920*). At the generic level, Nematodes were identified using keys provided by *Warwick, Platt & Somerfield (1998)* and *Decraemer & Smol (2006)*. At a specific level, the identification was performed by comparing the characteristics of the species found with those mentioned in the original descriptions. Drawings and photographs were produced utilizing an Olympus CX 31 optical microscope equipped with a drawing tube. Body measurements were obtained with the assistance of a mechanical map meter.

For scanning electron microscopy (SEM), individuals were obtained by disassembling glycerin-paraffin slides. These specimens were rehydrated with distilled water according to the method described by *Abolafia (2015)*. The individuals were then placed in a meiofauna processing manufactured container described by *Abolafia (2015)*, gradually dehydrated in a graded ethanol series (10% during one day and 20, 30, 40, 50, 60, 70, 80, 90, 92, 95, first 100 and second 100% along the second day, changing from one concentration to the following every 2 h) and dried in a critical-point dryer. The specimens were removed from the container, deposited on an aluminum stub covered with conductive tape, coated with gold and examined under a TM4000 SEM at 10 kV with BSE detector.

The holotype and one female paratype for each species are deposited in the Nematoda Collection at the Museum of Oceanography Prof. Petronio Alves Coelho (MOUFPE) in Brazil. Additional paratypes are deposited in the Meiofauna Laboratory within the Zoology Department at the Federal University of Pernambuco (NM LMZOO-UFPE).

The electronic edition of this article, formatted in Portable Document Format (PDF), constitutes a published study in accordance with the guidelines set forth by the International Commission on Zoological Nomenclature (ICZN). Consequently, the new names presented in this electronic version are considered effectively published under the Code based solely on the electronic edition. This research, along with its nomenclatural actions, has been duly registered in ZooBank, the ICZN's online registration system. The ZooBank LSIDs (Life Science Identifiers) can be accessed and the related information can be viewed using any standard web browser by appending the LSID to the prefix http://zoobank.org/. The LSID for this publication is: urn:lsid:zoobank.org:pub:01681568-8A01-47FC-96E5-DE43C7529F14. The online version of this research is preserved and can be accessed through the following digital repositories: PeerJ, PubMed Central, and CLOCKSS.

# RESULTS

## Systematics

**Taxonomic classification, after** *De Ley, Decraemer & Eyualem (2006)*
**Class Chromadorea** *Inglis, 1983*
**Subclass Chromadoria** *Pearse, 1942*
**Order Desmodorida** *De Coninck, 1965*
**Suborder Desmodorina** *De Coninck, 1965*
**Superfamily Desmodoroidea** *Filipjev, 1922*
**Family Desmodoridae** *Filipjev, 1922*
**Subfamily Molgolaiminae** *Jensen, 1978*
**Genus** *Molgolaimus  Ditlevsen, 1921*

**Diagnosis.** (After *Leduc, Fu & Zhao, 2019*) Cuticle finely striated, but may appear smooth under light microscopy (**prominently annulated in M. pecticauda**). Inner and outer labial papillae are small and often challenging to differentiate. Cephalic setae positioned slightly anterior or posterior to the head constriction. Amphidial fovea round and situated posterior to the head constriction. Buccal cavity small and weakly cuticularized, **featuring small teeth that may sometimes be indistinct**. Pharynx cylindrical, narrow and characterized by a prominent posterior bulb, typically spherical. Pharyngeal lumen weakly cuticularized, except within the pharyngeal bulb, where it may be heavily cuticularized. Secretory-excretory pore located anterior to the nerve ring and rarely found posterior to it. Female didelphic-amphidelphic, with reflexed ovaries (position of the genital branches varibles). Male monorchic with single anterior testis (**variable position of the genital branch: on the right, on the left or positioned ventrally in relation to the intestine**). Spicules of variable length and shape. Gubernaculum **present or absent, when present** with or without apophysis. Precloacal supplements often observed. Tail of varying shape and length (short and conical to elongated and conico-cylindrical).
**Type species:** *Molgolaimus tenuispiculum Ditlevsen, 1921*

## Valid species (Table 2)

The valid species were divided into groups and, when relevant, into subgroups following the classification by *Fonseca, Vanreusel & Decraemer (2006)* with modifications in subgroups 1b$_1$, 1b$_2$, 4a and 4b (Table 2). The list of valid species is in accordance with *Leduc, Fu & Zhao (2019)*, but with the addition of species described later and species described by *Bussau (1993)*. The species described by *Bussau (1993)* are currently considered valid (*Holovachov, 2020*). Valid species for which synonyms occur are listed in Appendix S1.

**Table 2  Valid species of *Molgolaimus* and division of groups and subgroups species based on *Fonseca, Vanreusel & Decraemer (2006)* with modifications to the ordering criteria for subgroups 1b₁, 1b₂, 4a e 4b.** Not applicable (-); b = de Man's ratio (*De Man, 1880*); amph ant/hd = distance between the anterior edge of the amphidial fovea in relation to the anterior end of the body divided by the head diameter.

| Group | Subgroup | Valid species |
|---|---|---|
| **Group 1** (spicule length < 35 μm) | **Subgroup 1a** (spicules = 1 cloacal body diameters long) | *Molgolaimus brevispiculum* **sp. nov.** |
| | | *M. citrus Gerlach, 1959* |
| | | *M. cuanensis* (*Platt, 1973*) *Jensen, 1978* |
| | | *M. euryformis Zhou et al. 2020* |
| | | *M. haakonmosbiensis Portnova, 2009* |
| | | *M. lazonus* (*Vitiello, 1970*) *Jensen, 1978* |
| | | *M. parallgeni* (*Vitiello, 1973*) *Jensen, 1978* |
| | | *M. turgofrons* (*Lorenzen, 1971b*) *Jensen, 1978* |
| | **Subgroup 1b₁** (spicules > 1 and <3 cloacal body diameters long; amph ant/hd ≥ 2) | *M. amphimacrus Bussau, 1993* |
| | | *M. drakus Fonseca, Vanreusel & Decraemer, 2006* |
| | | *M. gazii Muthumbi & Vincx, 1996* |
| | | *M. mareprofundus Fonseca, Vanreusel & Decraemer, 2006* |
| | | *M. porosus Bussau, 1993* |
| | | *M. spirifer* (*Warwick, 1970*) *Shi & Xu, 2016* |
| | **Subgroup 1b₂** (spicules >1 and <3 cloacal body diameters long; amph ant/hd <2) | *M. abyssorum Muthumbi & Vincx, 1996* |
| | | *M. carpediem Fonseca, Vanreusel & Decraemer, 2006* |
| | | *M. exceptionregulum Fonseca, Vanreusel & Decraemer, 2006* |
| | | *M. falliturvisus Fonseca, Vanreusel & Decraemer, 2006* |
| | | *M. galluccii Fonseca, Vanreusel & Decraemer, 2006* |
| | | *M. kiwayui Muthumbi & Vincx, 1996* |
| | | *M. longicaudatus Sun & Huang, 2024* |
| | | *M. minutus Jensen, 1988* |
| | | *M. pecticauda* (*Murphy, 1966*) *Shi & Xu, 2016* |
| | | *M. sapiens Fonseca, Vanreusel & Decraemer, 2006* |
| | | *M. sigmoides* **sp. nov.** |
| | **Subgroup 1c** (spicules > 3 cloacal body diameters long) | *M. typicus Furstenberg & Vincx, 1992* |
| | | *M. tyroi Muthumbi & Vincx, 1996* |
| **Group 2** (spicule length 35–53 μm) | - | *M. allgeni* (*Gerlach, 1950*) *Jensen, 1978* |
| | | *M. australis Fonseca, Vanreusel & Decraemer, 2006* |
| | | *M. macilenti Fonseca, Vanreusel & Decraemer, 2006* |
| | | *M. nettoensis Fonseca, Vanreusel & Decraemer, 2006* |
| | | *M. sabakii Muthumbi & Vincx, 1996* |
| | | *M. xuxunaraensis Fonseca, Vanreusel & Decraemer, 2006* |
| **Group 3** (spicule length 53–80 μm) | - | *M. liberalis Fonseca, Vanreusel & Decraemer, 2006* |
| | | *M. unicus Fonseca, Vanreusel & Decraemer, 2006* |
| | | *M. walbethi Fonseca, Vanreusel & Decraemer, 2006* |
| **Group 4** (spicule length >80 μm) | **Subgroup 4a** (b = 8–11; spicules = 4–6 cloacal body diameters long) | *M. gigaslongincus Fonseca, Vanreusel & Decraemer, 2006* |
| | | *M. kaikouraensis Leduc, Fu & Zhao, 2019* |
| | | *M. pacificus Fonseca, Vanreusel & Decraemer, 2006* |
| | | *M. tenuispiculum Ditlevsen, 1921* |
| | **Subgroup 4b** (species not included in subgroup 4a) | *M. gigasproximus Fonseca, Vanreusel & Decraemer, 2006* |
| | | *M. longispiculum* (*Timm, 1961*) *Jensen, 1978* |
| | | *M. paralongispiculum* **sp. nov.** |
| | | *M. tanai Muthumbi & Vincx, 1996* |

## Invalid species

The species listed below are considered *taxon inquirendum* for being poorly described (*Jensen, 1978*).

*Molgolaimus labradorensis* (*Allgén, 1957*) *Jensen, 1978*
*Molgolaimus tenuicaudatus* (*Allgén, 1959*) *Jensen, 1978*
*Molgolaimus tenuilaimus* (*Allgén, 1932*) *Jensen, 1978*

## Description of new species

*Molgolaimus sigmoides* **sp. nov.**
(Table 3; Figs. 1–2)

**Type material**. Four males and two females found. Holotype male (MOUFPE 0026), paratype female 1 (MOUFPE 0027), 3 male paratype (493–495 NM LMZOO-UFPE) and paratype female 2 (496–497 NM LMZOO-UFPE).

**Type locality**. South Atlantic Ocean, Continental shelf of the State of Bahia, Brazil, station 23 (S13°04′10.32″ W38°25′46.98″), December 11, 2019, 65 m.

**Locality of paratypes**. Paratype female 1: South Atlantic Ocean, Continental shelf of the State of Rio Grande do Norte, Brazil, station 2 (S05°42′54.42″ W34°59′31.92″), November 28, 2019, 60 m. Paratype males (1–3): South Atlantic Ocean, Continental shelf of the State of Sergipe, Brazil, station 17 (S11°00′00.54″ W36°49′58.98″), December 12, 2019, 54 m. Paratype female 2: South Atlantic Ocean, Continental shelf of the State of Sergipe, Brazil, station 16 (S10°44′59.28″ W36°25′32.88″), December 09, 2019, 58 m.

**Etymology**. The spicules of *Molgolaimus sigmoides* **sp. nov.** has a S-shaped structure. In Greek, sigma ($\varsigma$) corresponds to the letter S in the Latin alphabet.

**Holotype male**. Body cylindrical 376.5 μm long. Maximum body diameter corresponding to three times the head diameter. Cuticle faintly striated. Somatic setae not observed. Head slightly set off from the rest of the body by a slight constriction. Inner and outer labial sensilla indistinct. Four short cephalic setae (<two μm long) located at the level of buccal cavity and slightly anterior to head constriction (three μm from anterior end). Amphidial fovea circular, located 10 μm from anterior end (1.7 times the head diameter) and occupying 53% of corresponding body diameter. Buccal cavity small, narrow, with lightly slightly cuticularized walls. Three small teeth, difficult to see (a slightly larger dorsal tooth and two smaller ventrosublateral). Pharynx muscular (73 μm long), surrounding buccal cavity, consisting of narrow, cylindrical anterior portion and with conspicuous spherical posterior bulb (81% of corresponding body diameter). Pharyngeal lumen slightly cuticularized, except in the pharyngeal bulb where the valves are more cuticularized. Nerve ring situated at 59% of the pharynx length from anterior end. Secretory-excretory pore not observed. Ventral gland located posterior to the pharyngo-intestinal junction. Cardia

**Table 3 Morphometric data of *Molgolaimus sigmoides* sp. nov.** The measurements are expressed in micrometers, or if noted, as a percentage or ratio. Not applicable (*); a, b, c, c' = de Man's ratios (*De Man, 1880*).

| *Molgolaimus sigmoides* sp. nov. | Holotype | Male paratypes (*n* = 3) | Paratype female 1 | Paratype female 2 |
|---|---|---|---|---|
| Body length | 376.5 | 357–368 | 397 | 390 |
| Cephalic setae length | <2 | <2 | <2 | <2 |
| Head diameter | 6 | 4.5–5.5 | 5 | 5 |
| Distance from anterior end to cephalic setae | 3 | 2.5–3 | 3 | 3 |
| Distance from anterior end to amphidial fovea | 10 | 7–9 | 10 | 8.5 |
| Distance from anterior end to amphidial fovea in relation to head diameter | 1.7 | 1.5–1.7 | 2 | 1.7 |
| Amphidial fovea diameter (maximum width) | 4.5 | 4.5–4.5 | 3.5 | 4 |
| Body diameter at level of the amphidial fovea | 8.5 | 8–9 | 9 | 9 |
| % of the amphidial fovea diameter in relation to corresponding body diameter | 53% | 50–58% | 39% | 44% |
| Pharynx length | 73 | 65.5–68.5 | 68.5 | 70 |
| Position of nerve ring from anterior end | 43 | 39–42 | 40 | 42 |
| Nerve ring position in relation to pharynx length (%) | 59% | 57–63% | 58% | 60% |
| Pharyngeal bulb diameter | 13 | 12–13 | 11 | 14 |
| Body diameter at level of the pharyngeal bulb | 16 | 15–16 | 13 | 17 |
| % of basal bulb diameter in relation to corresponding body diameter | 81% | 80–85% | 85% | 82% |
| Maximum body diameter | 18 | 16 | 15 | 20.5 |
| Anal or cloacal body diameter | 15 | 14–14.5 | 10 | 11.5 |
| Tail length | 56 | 51–54 | 65 | 54 |
| Length of spicules along arc | 30 | 29–31.5 | * | * |
| Length of spicules along cord | 17.5 | 16–18 | * | * |
| Length of gubernaculum | 5 | 4.5–6 | * | * |
| Width of gubernaculum | 4 | 4 | * | * |
| Length of apophysis | 5 | 4.5–7 | * | * |
| Length of spicules along arc in relation to cloacal body diameter | 2 | 2–2.2 | * | * |
| Distance from anterior end to vulva | * | * | 170 | 175.5 |
| Position of vulva from anterior end (%) | * | * | 43% | 45% |
| Body diameter in vulva region | * | * | 15 | 19.5 |
| Anterior ovary length | * | * | 105 | 69 |
| Posterior ovary length | * | * | 107 | 70 |
| Reproductive system length | 200 | 181–211.5 | 118.5 | 139 |
| % of reproductive system in relation to body length | 53% | 50–59% | 30% | 36% |
| a | 21 | 23 | 26.5 | 19 |
| b | 5 | 5–5.5 | 6 | 6 |
| c | 7 | 7 | 6 | 7 |
| c' | 4 | 4 | 6.5 | 5 |

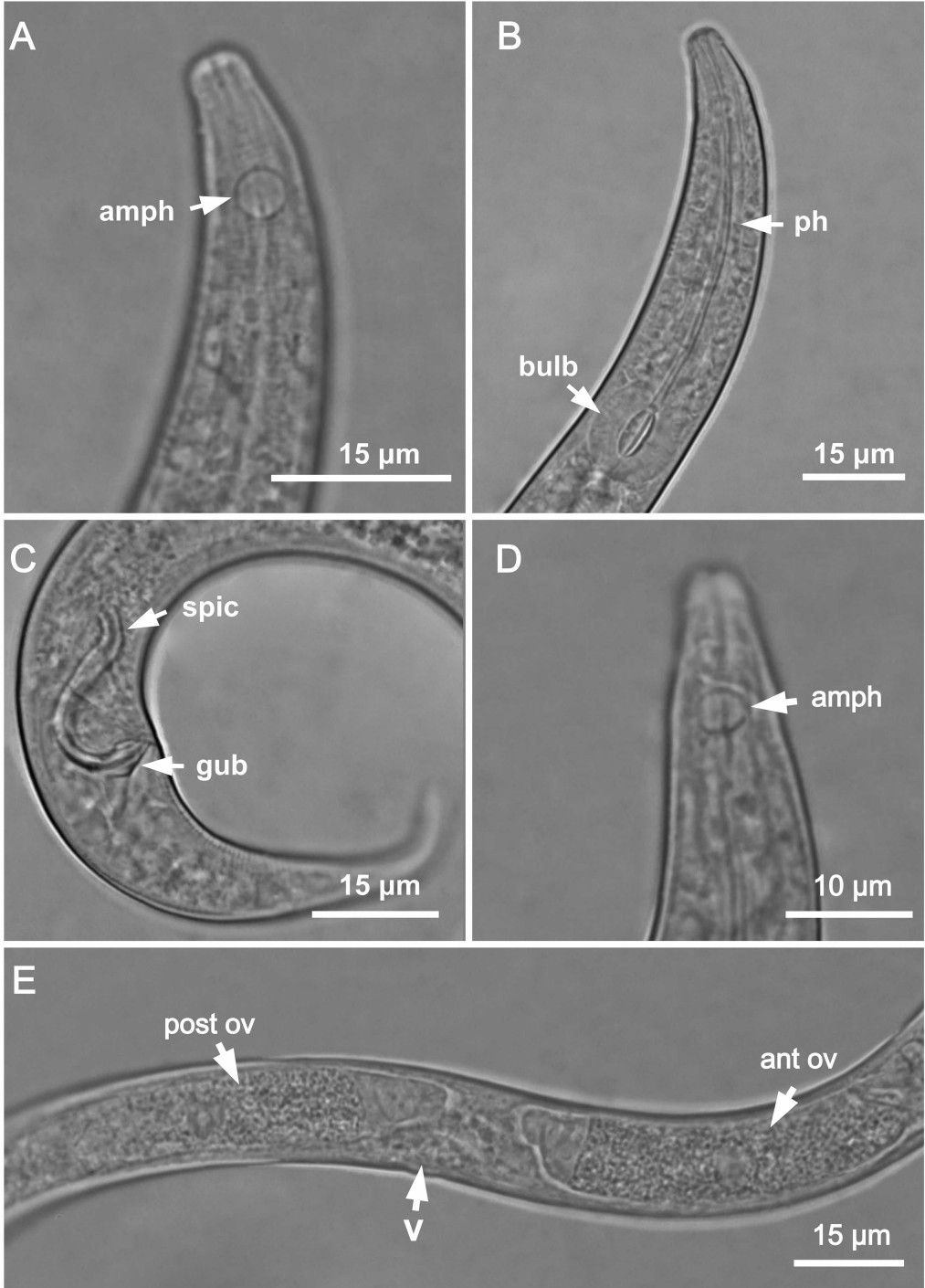

**Figure 1** *Molgolaimus sigmoides* **sp. nov. Holotype male, paratype female 1 and paratype female 2.** Holotype male: (A) anterior end (arrow indicating amphidial fovea = amph), (B) anterior region (arrows indicating pharynx = ph and basal bulb = bulb), (C) posterior end (arrows indicating spicule = spic and gubernaculum = gub). Paratype female 2: (D) anterior region (arrow indicating amphidial fovea = amph). Paratype female 1: (E) reproductive system (arrows indicating vulva = V; anterior ovary = ant ov and posterior ovary = post ov).

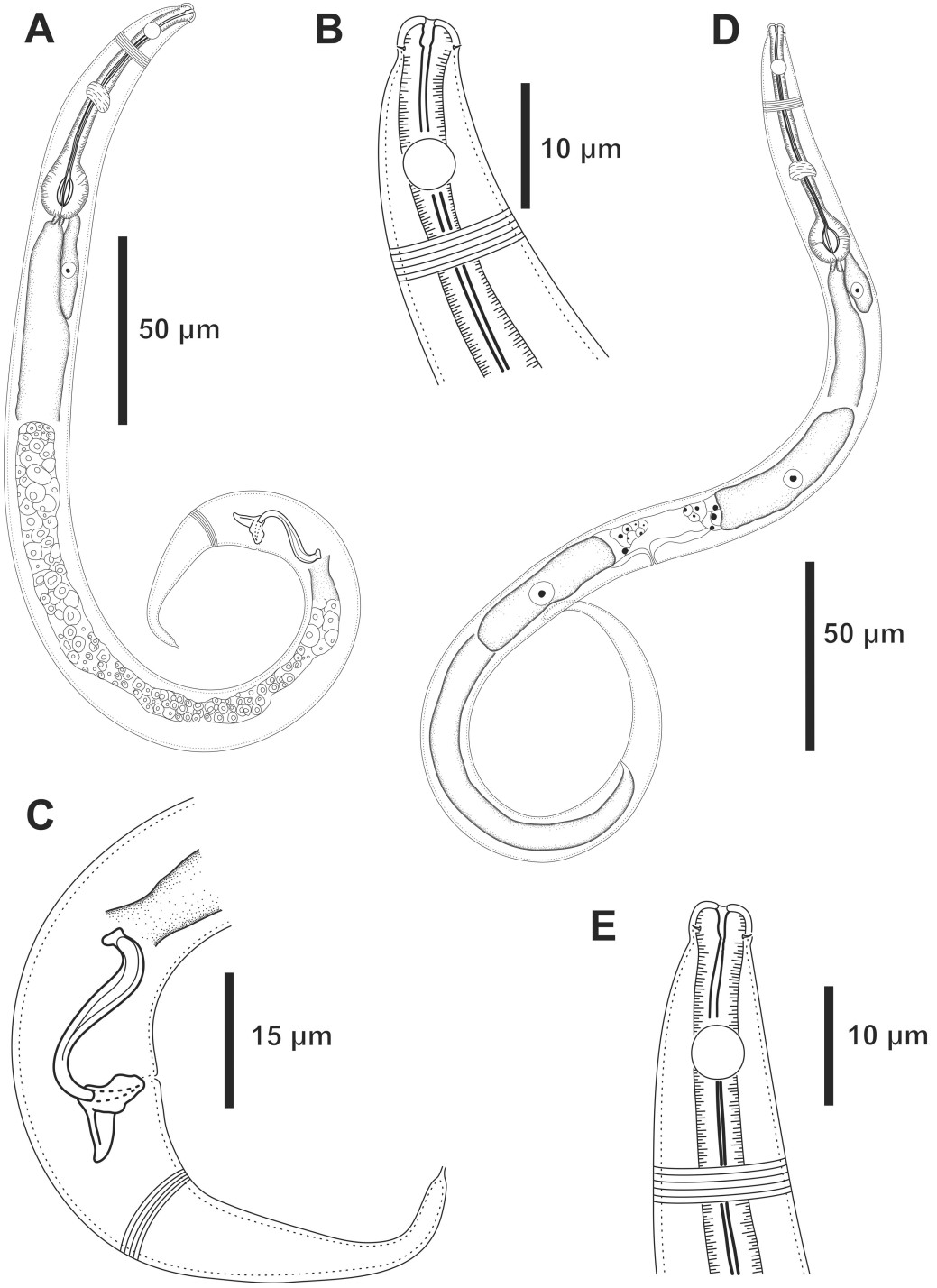

**Figure 2** *Molgolaimus sigmoides* **sp. nov. Holotype male and paratype female 1.** Holotype male: (A) overview, (B) anterior end, (C) posterior region. Paratype female 1: (D) overview, (E) anterior end.

partially surrounded by intestine. Reproductive system with single anterior outstretched testis to the left of the intestine. Spicules S-shaped, arched ventrally at the anterior end and dorsally at the posterior end (2 times the cloacal body diameter). Gubernaculum surrounding the spicules at the distal end and with dorsal-caudal apophyses. Precloacal supplements absent. Caudal glands not observed. Tail conico-cylindrical, about four times the cloacal body diameter. The cylindrical portion of the tail represents about 18% of its total length. Spinneret short.

**Paratype female**. Similar to male. Body measuring 397 μm in length, with a maximum diameter of 15 μm (three times the head diameter). Amphidial fovea, occupying 39% of corresponding body width and located 10 μm from anterior end. Basal bulb occupying 85% of the corresponding body diameter. Nerve ring situated at 58% of the pharynx length, from anterior end. Vulva located 170 μm from anterior end, at 43% of body length. Reproductive system didelphic, with reflexed ovaries. Anterior gonad situated to the right side of the intestine, posterior gonad to the left side of the intestine. Tail conico-cylindrical, about 6.5 times the anal body diameter. The cylindrical portion of the tail represents about 24% of its total length.

**Diagnosis**. *Molgolaimus sigmoides* **sp. nov.** characterized by its body length (357–397 μm). Cuticle finely annulated. Head slightly set off. Four short cephalic setae (<two μm long) located at the level of buccal cavity and slightly anterior to head constriction. Amphidial fovea occupying 50–58% of the corresponding body diameter in the males and 39–44% in the female, located at about 1.5–2 times the head diameter. Buccal cavity with three small teeth, one dorsal and two ventrosublateral, the dorsal is slightly larger. Muscular pharynx with conspicuous spherical posterior bulb (80–85% of the corresponding body diameter). Pharyngeal lumen slightly cuticularized, except in the pharyngeal bulb where the valves are more cuticularized. Spicules S-shaped (2–2.2 times the cloacal body diameter). Gubernaculum surrounding the spicules at the distal end and with dorsal-caudal apophysis. Tail conico-cylindrical which corresponds to 4–6.5 times the cloacal or anal body diameter.

**Differential diagnosis**. *Molgolaimus sigmoides* **sp. nov.** resembles *Molgolaimus sapiens* *Fonseca, Vanreusel & Decraemer, 2006* and *Molgolaimus xuxunaraensis* *Fonseca, Vanreusel & Decraemer, 2006* mainly due to the peculiar shape of the spicules (S-shaped), precloacal supplements absence and by present short cephalic setae (two μm long in *M. sapiens*; <two μm long in *M. sigmoides* **sp. nov.** and not see in *M. xuxunaraensis*). However, *M. sigmoides* **sp. nov.** it is the only species of *Molgolaimus* that has a gubernaculum with a dorsal-caudal apophysis. This characteristic differentiates the new species both from those morphologically closest to it (*M. sapiens* and *M. xuxunaraensis*) and from the other valid species of the genus.

*Molgolaimus paralongispiculum* **sp. nov.**
(Table 4; Figs. 3–5)

**Type material**. Five males and two females found. Holotype male (MOUFPE 0028), paratype female 1 (MOUFPE 0029), 4 male paratype (498–501 NM LMZOO-UFPE) and paratype female 2 (502–503 NM LMZOO-UFPE).

**Type locality**. South Atlantic Ocean, Continental shelf of the State of Pernambuco, Brazil, station 9 (S08°38′20.46″ W34°45′45.12″), November 26, 2019, 47 m.

**Locality of other paratypes**. South Atlantic Ocean, Continental shelf of the State of Alagoas, Brazil, station 11 (S09°15′30.54″ W34°57′13.14″), November 26, 2019, 87 m.

**Etymology**. The species name refers to similarity with species *Molgolaimus longispiculum* (*Timm, 1961*) *Jensen, 1978*.

**Holotype male**. Body cylindrical 811.5 µm long. Maximum body diameter corresponding about 4 times the head diameter. Cuticle faintly striated. Small papillae present throughout the body without a distribution pattern (observed in Paratypes 3 and 4, SEM photographs). Head off from the rest of the body by a constriction. Inner and outer labial sensilla indistinct. Four setiform cephalic setae (about 0.4 times the head diameter) located behind the head constriction (about six µm from anterior end). Amphidial fovea circular, located nine µm from anterior end (about 1.3 times the head diameter) and occupying 43% of corresponding body diameter. A small papilla associated with a cuticular pore present on both lateral edges of each amphidial fovea (distinctly visible in Paratypes 3 and 4, SEM photographs). Buccal cavity small, narrow, with slightly cuticularized walls. Three small teeth, a slightly larger dorsal tooth and two smaller ventrosublateral. Pharynx muscular (107.5 µm long), surrounding buccal cavity, consisting of narrow, cylindrical anterior portion and with conspicuous pyriform posterior bulb (78% of corresponding body diameter). Pharyngeal lumen slightly cuticularized, except in the pharyngeal bulb where the valves are more cuticularized. Nerve ring situated at 65% of the pharynx length from anterior end. Secretory-excretory pore not observed. Ventral gland located posterior of the pharyngo-intestinal junction. Cardia partially surrounded by intestine. Reproductive system with single anterior outstretched testis located ventrally to the intestine (germinal zone slightly to the right of the intestine). Spicules thin and elongated (6.9 times the cloacal body diameter). Gubernaculum funnel-shaped surrounding the spicules at the distal end and with anteriorly oriented apophysis. Precloacal supplements absent. Three caudal glands. Tail conico-cylindrical, about five times the cloacal body diameter. The cylindrical portion of the tail represents about one third of its total length. Spinneret short.

**Paratype female**. Similar to male. Body measuring 790.5 µm in length, with a maximum diameter of 36 µm (about five times the head diameter). Cephalic setae corresponding 0.5 times the head diameter. Amphidial fovea occupying 55% of corresponding body width and located nine µm from anterior end. Basal bulb occupying 80% of the corresponding body diameter. Nerve ring situated at 64% of the pharynx length, from anterior end. Secretory-excretory pore and ventral gland not observed. Vulva located 367.5 µm from anterior end, at 46% of body length. Reproductive system didelphic, with reflexed ovaries. Anterior gonad situated to the left side of intestine, posterior gonad to the right side of intestine. Tail conical with cylindrical terminal portion, about four times the anal body diameter. The cylindrical portion of the tail represents about 38% of its total length. Spinneret short.

Table 4 **Morphometric data of *Molgolaimus paralongispiculum* sp. nov.** The measurements are expressed in micrometers, or if noted, as a percentage or ratio. Not applicable (*); a, b, c, c' = de Man's ratios (*De Man, 1880*).

| *Molgolaimus paralongispiculum* sp. nov. | Holotype | Male paratypes ($n = 4$) | Paratype female 1 | Paratype female 2 |
|---|---|---|---|---|
| Body length | 811.5 | 693–778.5 | 790.5 | 732 |
| Cephalic setae length | 3 | 3–3.5 | 3.5 | 2.5 |
| Head diameter | 7 | 5–7 | 7 | 7 |
| Distance from anterior end to cephalic setae | 6 | 5–6 | 6 | 6 |
| Cephalic setae in relation to head diameter (%) | 43% | 43%–50% | 50% | 36% |
| Distance from anterior end to amphidial fovea | 9 | 6.5–11 | 9 | 8 |
| Distance from anterior end to amphidial fovea in relation to head diameter | 1.3 | 0.9–1.8 | 1.3 | 1.1 |
| Amphidial fovea diameter (maximum width) | 5.5 | 5–6 | 6 | 6 |
| Body diameter at level of the amphidial fovea | 10 | 8–10 | 11 | 10 |
| % of the amphidial fovea diameter in relation to corresponding body diameter | 55% | 60% | 55% | 60% |
| Pharynx length | 107.5 | 101.5–109 | 110.5 | 97 |
| Position of nerve ring from anterior end | 70 | 62–68 | 70.5 | 61 |
| Nerve ring position in relation to pharynx length (%) | 65% | 60–62% | 64% | 63% |
| Pharyngeal bulb diameter | 21 | 19–21.5 | 21.5 | 19 |
| Body diameter at level of the pharyngeal bulb | 27 | 23.5–28 | 27 | 25 |
| % of basal bulb diameter in relation to corresponding body diameter | 78% | 76–83% | 80% | 76% |
| Maximum body diameter | 33 | 28–35 | 36 | 31 |
| Anal or cloacal body diameter | 20.5 | 20.5–22 | 18 | 20 |
| Tail length | 96 | 74.5–98.5 | 80.5 | 82 |
| Length of spicules along arc | 140.5 | 134–145 | * | * |
| Length of spicules along cord | 94 | 109–125 | * | * |
| Length of spicules along arc in relation to cloacal body diameter | 6.9 | 6.1–7.1 | * | * |
| Length of gubernaculum | 19 | 19–20.5 | * | * |
| Length of gubernaculum in relation to length of spicules along arc (%) | 14% | 13–14% | * | * |
| Distance from anterior end to vulva | * | * | 367.5 | 334.5 |
| Position of vulva from anterior end (%) | * | * | 46% | 46% |
| Body diameter in vulva region | * | * | 36 | 31 |
| Anterior ovary length | * | * | 124.5 | 156 |
| Posterior ovary length | * | * | 174 | 136.5 |
| Reproductive system length | 480 | 391.5–439.5 | 237 | 214 |
| % of reproductive system in relation to body length | 59% | 53–57% | 32% | 29% |
| a | 25 | 22–27 | 22 | 24 |
| b | 8 | 6.5–8 | 7 | 8 |
| c | 8 | 8–9 | 10 | 9 |
| c' | 5 | 4–4.5 | 4 | 4 |

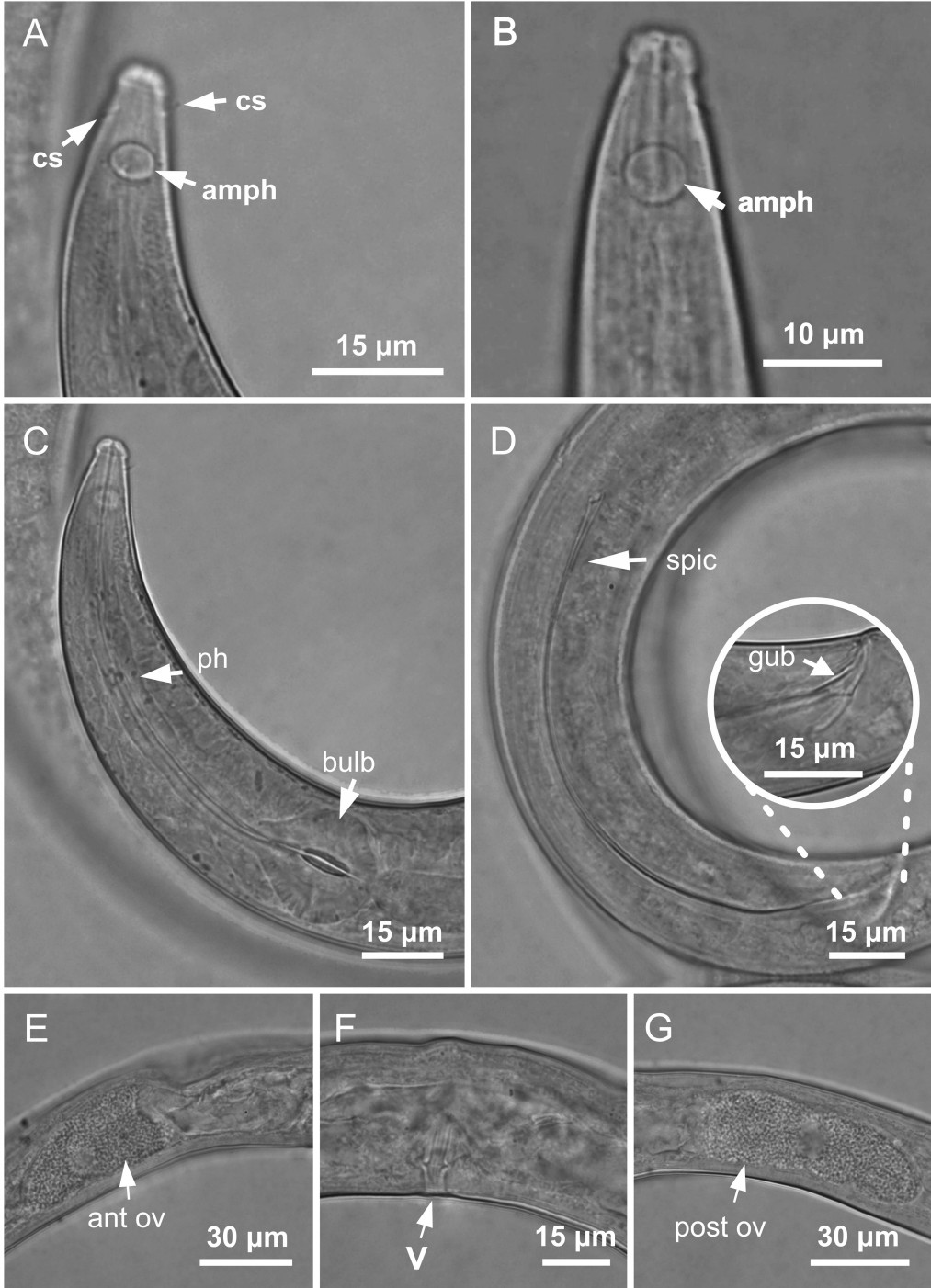

**Figure 3** *Molgolaimus paralongispiculum* **sp. nov. Holotype male and paratype female 1.** Holotype male: (A) anterior end (arrows indicating amphidial fovea = amph and cephalic setae = cs), (C) anterior region (arrows indicating pharynx = ph and basal bulb = bulb), (D) posterior end (arrow indicating spicule = spic and gubernaculum = gub). Paratype female: (B) anterior end (arrow indicating amphidial fove a= amph), (E) reproductive system (arrow indicating anterior ovary = ant ov), (F) reproductive system (arrow indicating vulva = V), (G) reproductive system (arrow indicating posterior ovary = post ov).

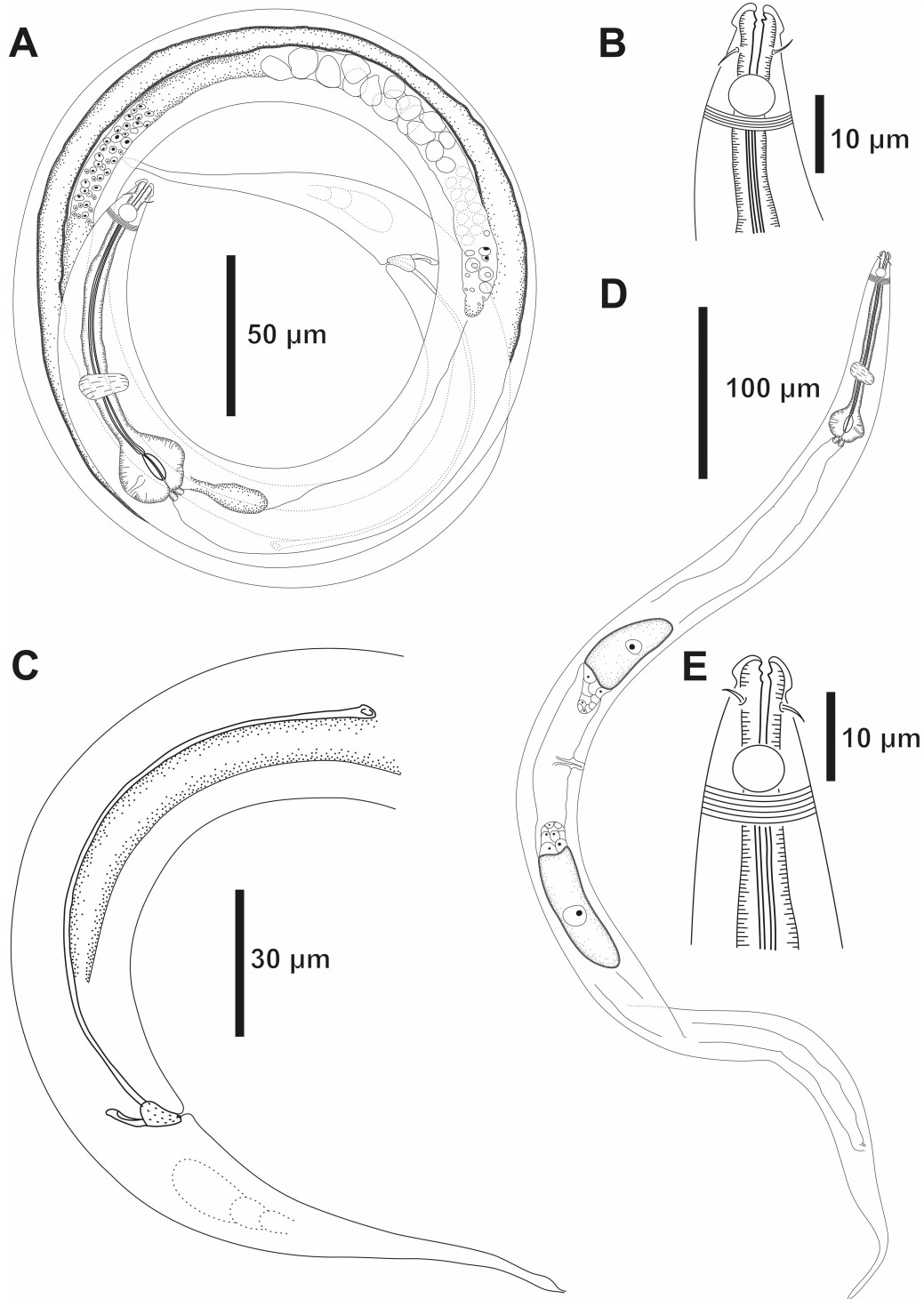

**Figure 4** *Molgolaimus paralongispiculum* **sp. nov. Holotype male and paratype female 1.** Holotype male: (A) overview, (B) anterior end, (C) posterior region. Paratype female 1: (D) overview, (E) anterior end.

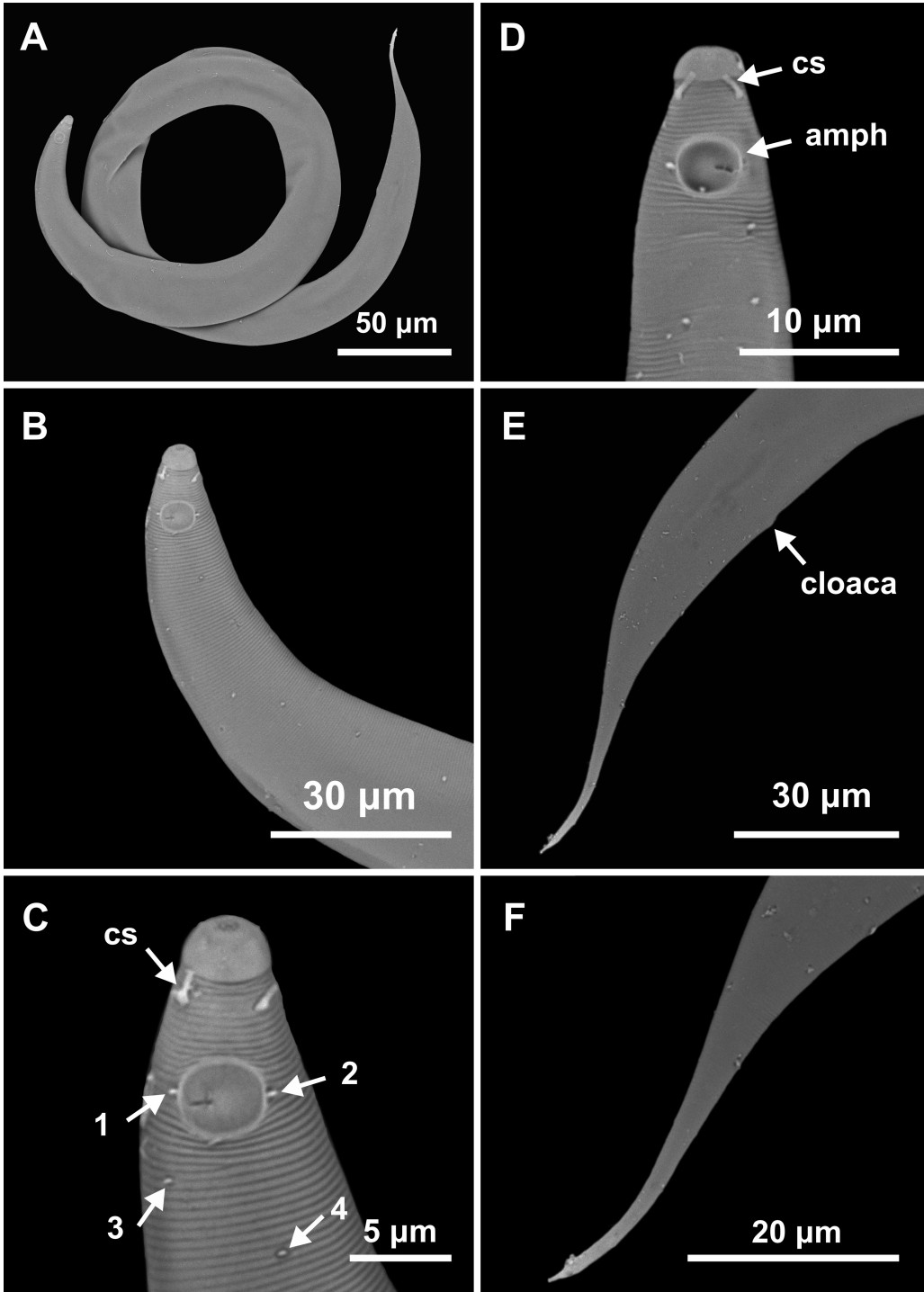

**Figure 5** *Molgolaimus paralongispiculum* **sp. nov. Male paratypes 3 and 4, SEM photographs.** Paratype 3: (A) overview, (B) anterior region, (C) anterior end (arrows indicating cephalic setae = cs and papillae positioned on both lateral edges of the amphidial fovea = 1 and 2; arrows indicating papillae found in the cervical region = 3 and 4); Paratype 4: (D) anterior end (arrows indicating amphidial fovea = amph and cephalic setae = cs); Paratype 3: (E) posterior region; (F) cylindrical portion of the tail.

**Diagnosis**. *Molgolaimus paralongispiculum* **sp. nov.** characterized by its body length (693–811.5 μm). Cuticle finely annulated. Head set off. Four setiform cephalic setae (about 0.4–0.5 times the head diameter) located behind the head constriction. Amphidial fovea occupying 55–60% of the corresponding body diameter, located at about 0.9–1.8 times the head diameter. Buccal cavity with three small teeth, one dorsal and two ventrosublateral, the dorsal is slightly larger. Muscular pharynx with conspicuous pyriform posterior bulb (76–83% of the corresponding body diameter). Pharyngeal lumen slightly cuticularized, except in the pharyngeal bulb where the valves are more cuticularized. Spicules thin and elongated (6.1–7.1 times the cloacal body diameter). Gubernaculum surrounding the spicules at the distal end and with anteriorly oriented apophysis. Tail conico-cylindrical (4–5 times the cloacal or anal body diameter).

**Differential diagnosis**. *Molgolaimus paralongispiculum* **sp. nov.** and four other species (*Molgolaimus longispiculum*; *Molgolaimus tenuispiculum* Ditlevisen, 1921; *Molgolaimus gigasproximus* Fonseca, Vanreusel & Decraemer, 2006 and *Molgolaimus gigaslongincus* Fonseca, Vanreusel & Decraemer, 2006) are the only representatives of the genus that have spicules >100 μm in length. Of these, *M. longispiculum* and *M. paralongispiculum* **sp. nov.** do not possess precloacal supplements, which helps to distinguish them from the other species mentioned above.

Considering only male specimens (female paratype described for *M. longispiculum* is immature), *M. paralongispiculum* **sp. nov.** shares the following features with *M. longispiculum*: de Man's ratios b (7.4 in *M. longispiculum* and 6.5–8 in the new species), c (8.2 in *M. longispiculum* and 8–9 in *M. paralongispiculum* **sp. nov.**) and c' (4.7 in *M. longispiculum* and 4–5 in *M. paralongispiculum* **sp. nov.**); total body length (737 μm long in *M. longispiculum* and 693–811.5 μm long in *M. paralongispiculum* **sp. nov.**) and the length of the cephalic setae (3 μm long in *M. longispiculum* and 3–3.5 μm long in *M. paralongispiculum* **sp. nov.**). However, the new species differs from *M. longispiculum* in relation to: position of the cephalic setae (cephalic setae positioned after the head constriction and below the level of the buccal cavity in *M. paralongispiculum* **sp. nov.** *vs* positioned close to the head constriction and at the same level of the buccal cavity in *M. longispiculum*); tail shape (conico-cylindrical in the new species *vs* conical with slightly swollen tip in *M. longispiculum*); de Man's ratio a (22–27 in *M. paralongispiculum* **sp. nov.** *vs* 35.5 in *M. longispiculum*) and the shape of the gubernaculum (surrounding the spicules at the distal end and with anteriorly oriented apophysis in the new species *vs* lamellar in *M. longispiculum*).

*Molgolaimus brevispiculum* **sp. nov.**
(Table 5; Figs. 6–7)

**Type material**. One male and one female found. Holotype male (MOUFPE 0030) and paratype female (MOUFPE 0031).

**Type locality**. South Atlantic Ocean, Continental shelf of the State of Bahia, Brazil, station 23 (S13°04′10.32″ W38°25′46.98″), November 12, 2019, 65 m.

**Locality of paratype female**. South Atlantic Ocean, Continental shelf of the State of Pernambuco, Brazil, station 10 (S08°56′36.78″W34°50″16.02″), November 26, 2019, 54 m.

**Etymology**. The specific epithet of the species name is due to its relatively short spicules length. Latin *brevis*: short in length.

**Holotype male**. Body cylindrical 465 µm long. Maximum body diameter corresponding to about 4 times the head diameter. Cuticle faintly striated. Somatic setae not observed. Head slightly set off from the rest of the body by a slight constriction. Inner and outer labial sensilla indistinct. Four cephalic setae (about 1.3 times the head diameter) located behind the head constriction (three µm from anterior end). Amphidial fovea circular, located 10 µm from anterior end (about 2.4 times the head diameter) and occupying 55% of corresponding body diameter. Buccal cavity tubular, narrow, with slightly cuticularized walls. Teeth not observed. Pharynx muscular (70.5 µm long), surrounding buccal cavity, consisting of narrow, cylindrical anterior portion and with conspicuous spherical posterior bulb (81% of corresponding body diameter). Pharyngeal lumen slightly cuticularized. Nerve ring situated at 66% of the pharynx length from anterior end. Secretory-excretory system short. Secretory-excretory pore located just below the nerve ring (about 72% of the pharynx length). Ventral gland located posterior to the pharyngo-intestinal junction. Cardia not observed. Reproductive system with single anterior outstretched testis to the left of the intestine. Short spicule, with slightly cephalized anterior end (1 time the cloacal body diameter). Gubernaculum and precloacal supplements absent. Tail conical, about 5 times the cloacal body diameter.

**Paratype female**. Similar to male. Body measuring 633 µm in length, with a maximum diameter of 17 µm (3.4 times the head diameter). Amphidial fovea, occupying 50% of corresponding body width and located 12 µm from anterior end. Basal bulb occupying 84% of the corresponding body diameter. Nerve ring situated at 65% of the pharynx length, from anterior end. Secretory-excretory pore located just below the nerve ring (about 68% of the pharynx length). Ventral gland located posterior to the pharyngo-intestinal junction. Vulva located 298 µm from anterior end, at 47% of body length. Reproductive system didelphic, with reflexed ovaries. Anterior ovary situated to the right side of the intestine, posterior ovary to the left side of the intestine. Tail conical, about six times the anal body diameter.

**Diagnosis**. *Molgolaimus brevispiculum* **sp. nov.** characterized by its body length (465–633 µm). Cuticle finely annulated. Head slightly set off. Four cephalic setae (0.9–1.3 times the head diameter) located behind the head constriction. Amphidial fovea occupying 50–55% of the corresponding body diameter, located at about 2.4 times the head diameter to the anterior end. Buccal cavity unarmed. Muscular pharynx with cuticularized lumen and conspicuous spherical posterior bulb (81–84% of the corresponding body diameter). Secretory-excretory system short. Secretory-excretory pore located just below the nerve ring. Short spicule, with slightly cephalized anterior end (one time the cloacal body diameter). Gubernaculum and precloacal supplements absent. Tail conical (five to six times the cloacal or anal body diameter).

**Differential diagnosis** (Table 6). Only males of *Molgolaimus typicus* (*Furstenberg & Vincx, 1992*) and *Molgolaimus drakus* (*Fonseca, Vanreusel & Decraemer, 2006*) have been

**Table 5** **Morphometric data of *Molgolaimus brevispiculum* sp. nov.** The measurements are expressed in micrometers, or if noted, as a percentage or ratio. Not applicable (*); a, b, c, c' = de Man's ratios (*De Man, 1880*).

| *Molgolaimus brevispiculum* sp. nov. | Holotype | Female paratype |
|---|---|---|
| Body length | 465 | 633 |
| Cephalic setae length | 5.5 | 4.5 |
| Distance from anterior end to cephalic setae | 3 | 2.5 |
| Head diameter | 4 | 5 |
| Cephalic setae in relation to head diameter (%) | 131% | 90% |
| Distance from anterior end to amphidial fovea | 10 | 12 |
| Distance from anterior end to amphidial fovea in relation to head diameter | 2.4 | 2.4 |
| Amphidial fovea diameter (maximum width) | 5.5 | 5 |
| Body diameter at level of the amphidial fovea | 10 | 10 |
| % of the amphidial fovea diameter in relation to corresponding body diameter | 55% | 50% |
| Pharynx length | 70.5 | 79.5 |
| Position of nerve ring from anterior end | 46.5 | 51.5 |
| Nerve ring position in relation to pharynx length (%) | 66% | 65% |
| Distance from anterior end to secretory-excretory pore | 50.5 | 54 |
| Pharyngeal bulb diameter | 13 | 16 |
| Body diameter at level of the pharyngeal bulb | 16 | 19 |
| % of basal bulb diameter in relation to corresponding body diameter | 81% | 84% |
| Maximum body diameter | 16 | 17 |
| Anal or cloacal body diameter | 13 | 11 |
| Tail length | 60 | 70 |
| Length of spicules along arc | 13 | * |
| Length of spicules along cord | 10.5 | * |
| Length of spicules along arc in relation to cloacal body diameter | 1.0 | * |
| Distance from anterior end to vulva | * | 298 |
| Position of vulva from anterior end (%) | * | 47% |
| Body diameter in vulva region | * | 17 |
| Anterior ovary length | * | 80.5 |
| Posterior ovary length | * | 69 |
| Reproductive system length | 306 | 106 |
| % of reproductive system in relation to body length | 66% | 17% |
| a | 29 | 37 |
| b | 7 | 8 |
| c | 8 | 9 |
| c' | 5 | 6 |

described, thus the comparison between those species and *Molgolaimus brevispiculum* **sp. nov.** is only based on males. The absence of gubernaculum is a common characteristic between *M. typicus*, *M. drakus* and *M. brevispiculum* **sp. nov.**, which are thus different

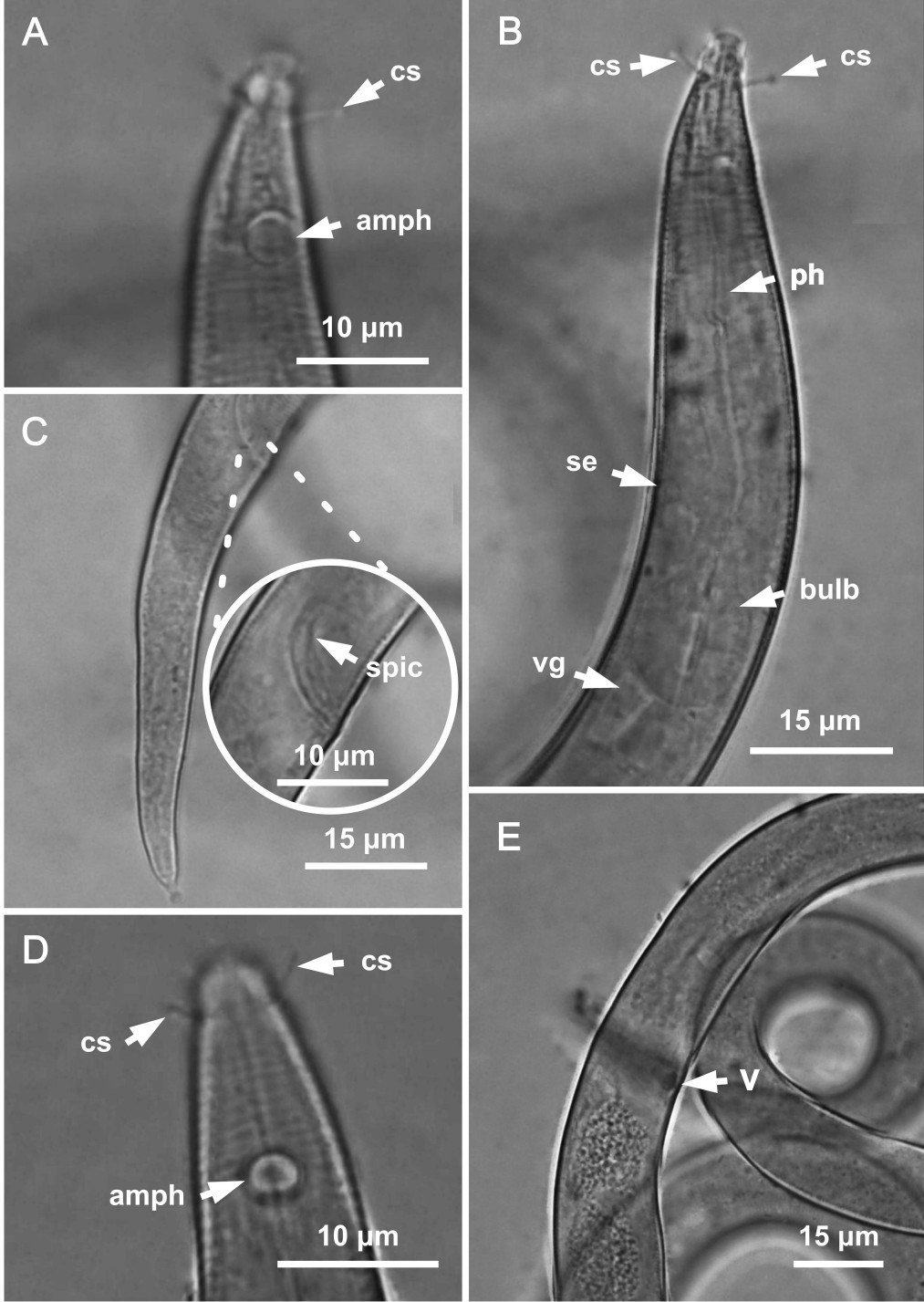

**Figure 6** *Molgolaimus brevispiculum* **sp. nov. Holotype male and paratype female.** Holotype male: (A) anterior end (arrows indicating cephalic setae = cs and amphidial fovea = amph), (B) anterior region (arrows indicating cephalic setae = cs; pharynx = ph; secretory-excretory pore = se; basal bulb = bulb and ventral gland = vg), (C) posterior end (arrow indicating spicule = spic). Paratype female: (D) anterior end (arrows indicating cephalic setae = cs and amphidial fovea = amph), (E) reproductive system (arrow indicating vulva = V).

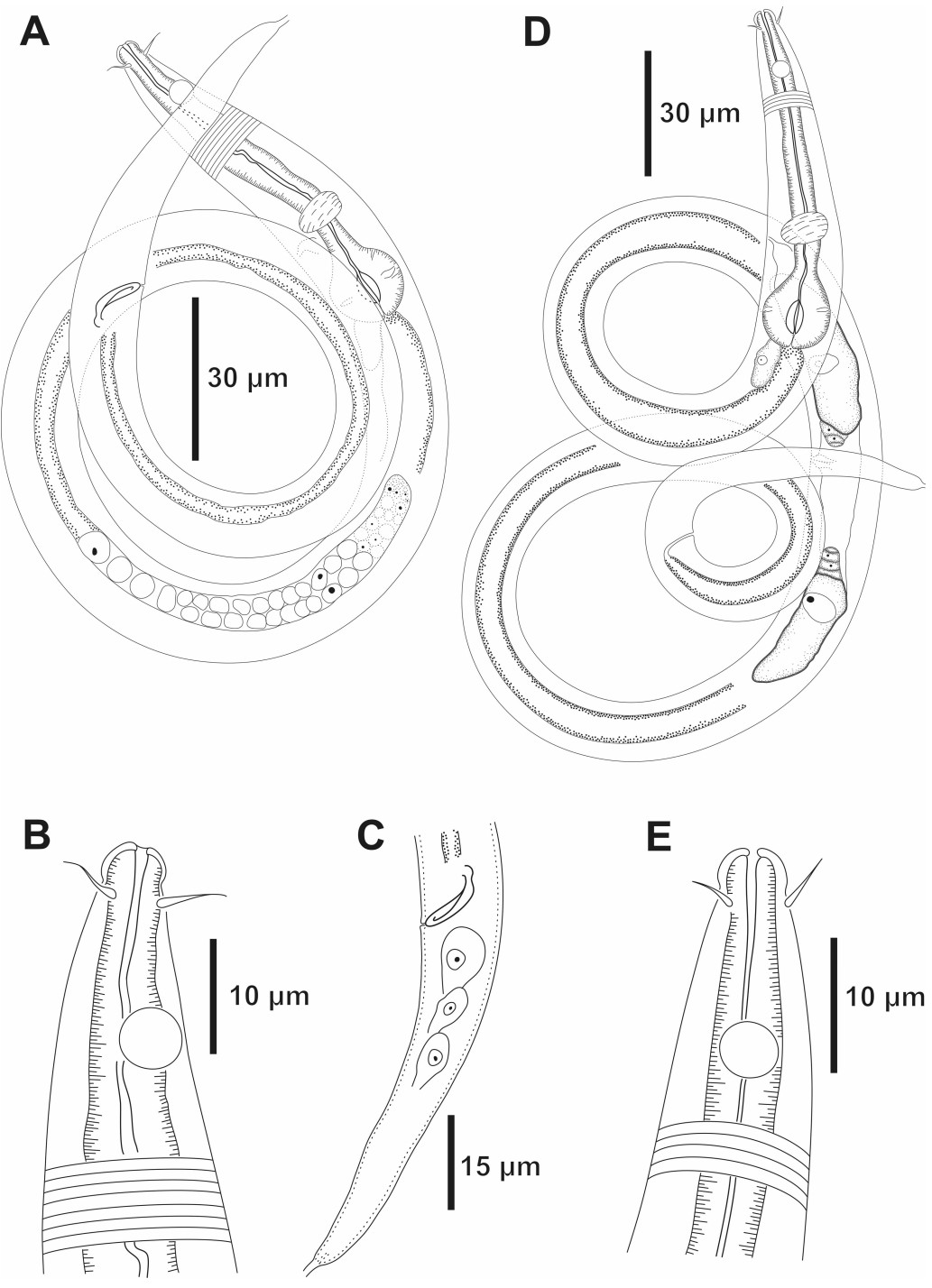

**Figure 7** *Molgolaimus brevispiculum* **sp. nov. Holotype male and paratype female.** Holotype male: (A) overview, (B) anterior end, (C) posterior region. Paratype female: (D) overview, (E) anterior end.

**Table 6 Comparison of species *Molgolaimus brevispiculum* sp. nov. with morphologically similar species (only males).** a, b, c, c' = de Man's ratios (*De Man, 1880*); (amph%) = percentage of the amphidial fovea diameter in relation to corresponding body diameter; (−) = parameter absent.

| | *M. typicus* | *M. drakus* | *M. brevispiculum* sp. nov. |
|---|---|---|---|
| Body length | 456 μm | 495–575 μm | 465 μm |
| a | 35 | 35.5–42.7 | 29 |
| b | 5.6 | 6.1–7 | 7 |
| c | 6.1 | 7–7.9 | 8 |
| c' | 6.8 | 5.3–6.2 | 5 |
| Cephalic setae length | 2 μm | 2 μm | 5.5 μm |
| amph% | 40% | 38–44% | 55% |
| Precloacal supplements | 2 small papillae | 1 small papillae | – |
| Spicules length | 30 μm | 19–20 μm | 13 μm |
| Gubernaculum | – | – | – |

from the other species of the genus *Molgolaimus*. Nonetheless, other features as the absence of precloacal supplements, the spicules length, the cephalic setae length, the size of the amphidial fovea in relation to the body diameter and the de Man's ratio help in differentiating those three species (Table 6). In addition, *M. brevispiculum* **sp. nov.** differs from *M. typicus* with regard to the position of the secretory-excretory pore, with this structure located at the same level as the amphidial fovea in *M. typicus* and just below the nerve ring in the new species.

## DISCUSSION

Even with the use of phylogenetic analyses based on SSU sequences to investigate the taxonomic position of *Molgolaimus*, this question remains unresolved. The two most recent phylogenetic studies involving species of this genus presented divergent results. The phylogenetic analyses performed by *Leduc, Fu & Zhao (2019)* suggest that *Molgolaimus* should be classified with the Chromadorida and not the Desmodorida or Microlaimida, while those performed by *Sun & Huang (2024)* indicate that the genus belongs to Desmodoridae (Desmodorida). These studies used different parameters to perform their respective analyses (*e.g.*, number of Orders and genus); *Sun & Huang (2024)* did not use SSU sequences of taxa belonging to Chromadorida, making it impossible to compare with the findings of *Leduc, Fu & Zhao (2019)*, thus making it implausible to confirm or refute their observations regarding the phylogenetic position of the genus. Similar to *Leduc, Verdon & Zhao (2018)*, previous studies investigated the phylogenetic position of *Molgolaimus* using the same tool and based on SSU sequences of *Molgolaimus demani* (*Meldal, 2004*; *Meldal et al., 2007*; *Cavalcante, 2010*; *Leduc & Zhao, 2016*), neglecting the fact that this species was transferred to the genus *Microlaimus* by *Lorenzen (1981)*, and is therefore considered a synonym of *Microlaimus tenuispiculum*. Considering this conflict and while the taxonomic position of *Molgolaimus* remains under investigation, here we adopt the classification provided by *De Ley, Decraemer & Eyualem (2006)*, which was based on both molecular evidence and morphological characteristics.

The *Molgolaimus* species grouping proposed by *Fonseca, Vanreusel & Decraemer (2006)*, based on the absolute length of the spicules is clear and helps in the separation/identification of species. However, the division of subgroups is not as simple, especially the division between subgroups $1b_1$ and $1b_2$. *Leduc, Fu & Zhao (2019)* listed the valid species of the genus following the division proposed by *Fonseca, Vanreusel & Decraemer (2006)*. When listing the species belonging to subgroup $1b_1$, they mentioned that this subgroup includes species whose spicules length measure between one and three times the cloacal body diameters. When listing the species of subgroup $1b_2$, they repeated the same information (species whose spicules lengths are equivalent to between one and three cloacal body diameters). The difference between subgroups $1b_1$ and $1b_2$ in the aforementioned list cannot be clearly identified. Furthermore, the order of presentation of group 4 (species with spicules lengths >80 $\mu$m), composed of subgroups 4a and 4b, does not follow a logical sequence. Subgroup 4a lists species that are not included in the grouping parameters of subgroup 4b (de Man's ratio b = 8–11, spicules = 4–6 cloacal body diameters long). However, by logical sequence, the species whose adopted criteria allow for the division of the subgroup, should be presented first and, subsequently, those that are not included in the adopted criteria. In order to make the separation and identification of *Molgolaimus* species more practical, we propose a modification of the division of subgroups $1b_1$ and $1b_2$ and the rearrangement of the order of presentation of subgroups 4a and 4b.

To modify the division of subgroups $1b_1$ and $1b_2$, we adopted the relative position of the amphidial fovea as a parameter, *i.e.,* the ratio of the distance between the anterior edge of the amphidial fovea in relation to the anterior end of the body divided by the head diameter (amph ant/hd) (Table 7). Although other characteristics can be used to designate the subgroups in question, this character is mentioned in most of the descriptions of *Molgolaimus* species. Furthermore, when absent in the description, it can be checked using the available images. The adopted proportion is used in other genera, such as *Microlaimus*, as part of a set of taxonomic tools to express similarity relationships or to identify differences between species (*Kovalyev & Tchesunov, 2005*; *Gagarin & Tu, 2014*; *Revkova, 2020*; *Lima, Neres & Esteves, 2022*; *Manoel, Neres & Esteves, 2024*). As in *Fonseca, Vanreusel & Decraemer (2006)*, the subgroups proposed here are only based on male individuals. As a criterion for separating subgroups, we used the ratio amph ant/hd described/measured from the holotype of each species. The variations found in the paratypes, when available, were indicated in parentheses in Table 7.

*Molgolaimus* species belonging to Group 1 that concomitantly have spicules lengths larger than one (>1) and smaller than three (<3) cloacal body diameters; ratio amph ant/hd greater than or equal to two ($\geq$2) are included in subgroup $1b_1$ (Table 7). Species that concomitantly have spicules lengths greater than one (>1) and less than three (<3) cloacal body diameters; ratio amph ant/hd less than two (<2) are included in subgroup $1b_2$ (Table 7). The species *M. exceptionregulum* (*Fonseca, Vanreusel & Decraemer, 2006*), *M. longicaudatus*, *M. pecticauda* (*Murphy, 1966*; *Shi & Xu, 2016*) and *M. sapiens* which were previously part of subgroup $1b_1$, now appear in subgroup $1b_2$. The species *M. amphimacrus* (*Bussau, 1993*) and *M. porosus* (*Bussau, 1993*), which are now considered valid (*Holovachov, 2020*), were placed in subgroup $1b_1$. The other species remained in their original subgroups.

**Table 7  Redistribution of *Molgolaimus species* into subgroups 1b$_1$ and 1b$_2$ by *Fonseca, Vanreusel & Decraemer (2006)*.** Information (measurements or proportions) solely based on males of the species Molgolaimus. Data obtained from the holotype of each species. Variations found in male paratypes, when available, were indicated in parentheses. Spicules length in relation to the cloacal body diameter = spic/cbd; distance of amphidial fovea from anterior end in relation to head diameter = amph ant/hd. Subgroup 1b$_1$: Molgolaimus species that concomitantly have ratio spic/cbd > 1 and < 3; amph ant/hd ≥ 2. Subgroup 1b$_2$: Molgolaimus species that concomitantly have ratio spic/cbd > 1 and < 3; amph ant/hd < 2.

| Species | Spicules length (µm) | Group | spic/cbd | amph ant/hd | Subgroup |
|---|---|---|---|---|---|
| *M. amphimacrus* | 34 | 1 | 2.4 | 3.5 | 1b$_2$... |
| *M. drakus* | 20 (19) | 1 | 1.5 | 2.4 | 1b$_1$ |
| *M. gazii* | 29 | 1 | 2 | 2.6 (3) | 1b$_1$ |
| *M. mareprofundus* | 32 (29) | 1 | 1.9 (1.6) | 2 (1.7) | 1b$_1$ |
| *M. porosus* | 25 | 1 | 1.9 | 2.8 | 1b$_1$ |
| *M. spirifer* | 25 (22–23) | 1 | 1.5 (1.2–1.3) | 3 (2.7–3) | 1b$_1$ |
| *M. abyssorum* | 20 (18–23) | 1 | 2 | <1 | 1b$_2$ |
| *M. carpediem* | 26 (23) | 1 | 1.6 (1.4) | 1.2 (1.25–1.5) | 1b$_2$ |
| *M. exceptionregulum* | 28 (29–31) | 1 | 1.6 (1.7–1.8) | 1.8 (2 –2.2) | 1b$_2$ |
| *M. falliturvisus* | 26 | 1 | 1.7 | 1.4 | 1b$_2$ |
| *M. gallucci* | 22 (22–27) | 1 | 1.7 (1.6–2.1) | 1.75 (1.8) | 1b$_2$ |
| *M. kiwayui* | 20 (22) | 1 | 1.7 | 1.5 (1) | 1b$_2$ |
| *M. longicaudatus* | 33 (32–35) | 1 | 1.8 (1.9) | 1 (1.2) | 1b$_2$ |
| *M. minutus* | 23 (25) | 1 | 2.1–1.9 | 1.4 | 1b$_2$ |
| *M. pecticauda* | 31.5 | 1 | 1.7 | 1.8 | 1b$_2$ |
| *M. sapiens* | 35 | 1 | 1.8 | 1.6 | 1b$_2$ |
| *M. sigmoides* **sp. nov.** | 30 (29–31.5) | 1 | 2 (2.1–2.2) | 1.7 (1.5–1.7) | 1b$_2$ |

In relation to group 4, following a logical sequence, subgroup 4b is now named subgroup 4a, which now includes species that simultaneously present: de Man's ratio b = 8–11, spicules = 4–6 cloacal body diameters long (see Table 2). Subgroup 4b is then intended for those species that are not included in the criteria mentioned above (see Table 2).

The new species *M. sigmoides* **sp. nov.** belongs to subgroup 1b$_2$ (spicules length >1 and <3 cloacal body diameter; ratio amph ant/hd <2). *Molgolaimus sigmoides* **sp. nov.** possess a gubernaculum with a dorsal-caudal apophysis, a characteristic never before reported for the genus. Mainly due to this, the new species can be confused and classified as belonging to the genus *Aponema Jensen, 1978*. However, it is important to note that females of all *Aponema* species possess a reproductive system with two opposite and outstretched ovaries. In *Molgolaimus*, the female reproductive system has two reflected ovaries. In appearance, the molgolaimids may be very similar to Microlaimidae but differ sharply with antidromously reflected ovaries (*Tchesunov, 2014*). Furthermore, representatives of the family Molgolaimidae have a slightly cuticularized pharynx associated with a pronounced pharyngeal bulb with sclerotized valves. This set of characteristics was not observed in *Aponema* species or in any representative of the family Microlaimidae *Micoletzky, 1922*.

*Molgolaimus paralongispiculum* **sp. nov.** belongs to subgroup 4b. Similarly to what was observed in *Molgolaimus tanai Muthumbi & Vincx, 1996*, *M. paralongispiculum* **sp. nov.**

possess testis ventrally positioned in relation to the intestine. In most species of the genus the male genital branch is positioned to the right or left of the intestine. Although it is not always possible to clearly visualize this structure, it is important to record the variation that exists within the genus. The variation in the position of the germinal branch in relation to the intestine was also added to the diagnosis of the genus.

*Molgolaimus brevispiculum* **sp. nov.** belongs to subgroup 1a (spicules = 1 cloacal body diameter long). In absolute numbers, *M. brevispiculum* **sp. nov.** is the species with the smallest spicules among the representatives of the genus. Additionally, the gubernaculum is absent in this species. The absence of the gubernaculum is a rare characteristic in the genus, having only been detected in two other species (*M. typicus* and *M. drakus*). The possibility of the presence or absence of the gubernaculum in species of *Molgolaimus* was added to the diagnosis of the genus.

Although the occurrence of the genus *Molgolaimus* has already been recorded for the Brazilian coast, none of these studies include the description of new species of the genus. This study presents three new species of *Molgolaimus* for the first time, originally described from sediments collected on the Continental Shelf break of Northeastern Brazil. This finding contributes significantly to knowledge about the richness of species of *Molgolaimus* found in the South Atlantic Ocean. Furthermore, we provide critical elements that lead to the reorganization of the division of *Molgolaimus* species within previously existing subgroups, thus facilitating the identification/differentiation of species within the genus. Considering that the taxonomic position of *Molgolaimus* remains uncertain, additional efforts should be made to resolve this issue.

## ACKNOWLEDGEMENTS

The authors thank the IAM-FIOCRUZ Core Facilities for the use of Electron Microscopy Services.

### Funding

This study was financed by the Coordenação de Aperfeiçoamento de Pessoal de Nível Superior—Brasil (CAPES)—Finance Code 001. The Brazilian navy provided logistical support for the scientific cruise aboard the R/V Vital de Oliveira. Alex Manoel was supported by a FACEPE graduate scholarship (IBPG-1516-2.00/21). The funders had no role in study design, data collection and analysis, decision to publish, or preparation of the manuscript.

### Grant Disclosures

The following grant information was disclosed by the authors:
Coordenação de Aperfeiçoamento de Pessoal de Nível Superior - Brasil (CAPES) - Finance Code 001.
The Brazilian navy.
FACEPE graduate scholarship: IBPG-1516-2.00/21.

## Competing Interests

The authors declare there are no competing interests.

## Author Contributions

- Alex Manoel conceived and designed the experiments, performed the experiments, analyzed the data, prepared figures and/or tables, authored or reviewed drafts of the article, and approved the final draft.
- Patrícia F. Neres conceived and designed the experiments, performed the experiments, analyzed the data, prepared figures and/or tables, authored or reviewed drafts of the article, and approved the final draft.
- Andre M. Esteves conceived and designed the experiments, performed the experiments, analyzed the data, prepared figures and/or tables, authored or reviewed drafts of the article, and approved the final draft.

## Data Availability

The data is available in the tables.

## New Species Registration

The following information was supplied regarding the registration of a newly described species:

Publication LSID: urn:lsid:zoobank.org:pub:01681568-8A01-47FC-96E5-DE43C7529F14

*Molgolaimus molgolaimus brevispiculum* species LSID: urn:lsid:zoobank.org:act:1A2F93FC-6321-4D0D-9F3A-408BA0896332;

*Molgolaimus molgolaimus paralongispiculum* species LSID: urn:lsid:zoobank.org:act: 3CF410BD-705B-4B95-A05B-05302349C63C;

*Molgolaimus molgolaimus sigmoides* species LSID: urn:lsid:zoobank.org:act:66445075-0CBB-4419-A450-6D0A79169918.

## Supplemental Information

Supplemental information for this article can be found online at http://dx.doi.org/10.7717/peerj.19156#supplemental-information.

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
