# Peer review of "First three new species of free-living marine nematodes of the Molgolaimus (Nematoda: Desmodoridae) from the continental shelf of the Brazilian coast (Atlantic Ocean)"

_PeerJ, doi:10.7717/peerj.19156_

## Round 0.1 · original submission · Minor Revisions

All reviewers agree on the high quality of the paper, as well as its originality and significance in advancing the taxonomy, biodiversity, and biogeography of the phylum Nematoda. I recommend addressing the reviewers’ suggestions and comments, revising the manuscript accordingly, and preparing a rebuttal letter to justify any changes you deem not appropriate.

Reviewer 1 ·

Basic reporting

Present study reports three new Molgolaimus species from the Brazilian coast with nice text descriptions and good illustrations.

I do not have specific comment on the manuscript for further improvement.

Experimental design

no comment

Validity of the findings

no comment

Reviewer 2 ·

Basic reporting

no comment

Experimental design

no comment

Validity of the findings

no comment

Additional comments

This is a proper taxonomic study containing descriptions of three new nematode species of the genus Molgolaimus, amended diagnosis of the genus and some rearrangement of species in groups of similarity.
The study is of good quality, I have only a few minor remarks.

Remarks:
Material and methods, Study area and sampling. It is necessary to describe the area and sampling here, the paper should be self-sufficient.
Laboratory processing. It is incomprehensible for me, were the samples fixed before extracting organisms or not? Were the nematodes transferred to formalin+glycerin alive? Please describe all the procedures in full.
Genus Molgolaimus Dilevsen, 1921. Diagnosis. ”Buccal cavity … with small teeth”. Please, denote number and position of the teeth.
Figure 2 B, E. Internal cuticular lining in the pharynx looks different in the preamphideal region (single-lined) and postamphideal region (double-lined). What does it mean – is there at amphid’s level a distinct border or some discontinuity between pre- and postamphideal lining? Then it should be depicted clearly.
Figures 5 and 7. Why are the body papillae shown on figs 7 C, D not depicted on the line drawings (fig. 4)? It should be done.

·

Basic reporting

Dear authors, congratulations on your work.
The manuscript submitted for publication in PeerJ presents the description of three new species of free-living marine nematodes of the genus Molgolaimus, a genus that has a high density in shelf environments.
It is a complete study, written in a clear and orderly manner, with a good presentation of the data. In my opinion it can be accepted for publication in PeerJ if some minor changes are made.
Introduction
It is necessary to talk a little more about the ecological characteristics of this genus and cover its occurrence, not only on shelf reefs in Brazil, but also in other shallower coastal habitats. Material and methods
Although the authors stated that the information regarding the study area and collection procedures were described in the article Manoel, Neres & Esteves (2024) Three new species of free-living marine nematodes of the Microlaimus genus (Nematoda: Microlaimidae) from the continental shelf off northeastern Brazil (Atlantic Ocean). I believe that it is essential to describe this in this article, since it is an important item for the work. In addition, I strongly suggest that the authors add a map of the bathymetry study area with information for better visualization and location of the platform from which the species were described. Since the location of the paratypes, the coordinates of the points and the name of the sampling station are placed.
Results and discussion
well described and organized

Experimental design

no comment

Validity of the findings

The manuscript submitted for publication in PeerJ presents the description of three new species of free-living marine nematodes of the genus Molgolaimus and an amendus of the diagnosis of the genus, a genus that has a high density in shelf environments.

Additional comments

NO additional comments

·

Basic reporting

Basic reporting
The paper the authors submitted is a description of three new species of the genus Molgolaimus from the continental shelf of the Brazilian coast. Globally, the paper is robust and detailed enough for publication, though some minor corrections and additions would improve the overall readability of the paper. The work the authors have put in for describing these three species is valuable and relevant to the scientific community, for taxonomists and for ecologists using biodiversity data.
The English used throughout the paper is clear, except for some minor mistakes. Concerning the introduction and the background, the history of the genus Molgolaimus is very well detailed with an interesting section on the molecular identification and placement of the genus. The structure conforms to PeerJ standards. The authors provided some good quality pictures of the specimens, which can be really helpful when identifying, together with some detailed and clear drawings. The authors also provide useful tables reporting the fundamental measurements required for the description and identification of the species. The SEM photographs are a nice additional element, which is not often provided in descriptions of new nematode species.
Concerning the tables, some authors (in other nematode species descriptions) include the averaged values of each morphological characteristic, as in a column for the holotype male, another one including the averaged values between all the male individuals and one with the averaged values between all the females. In this manuscript, the authors included the measurements of each individual, which is great, but for the reader and the taxonomist it may be interesting to have just one value (the average).
The description of new species of Molgolaimus is the core of the paper. Nonetheless, I believe more information about the ecological context could be helpful for the reader. I detailed this point further below in the review.
In addition, a simple figure of the study area would be appreciated, although not many papers of species description provide it. I believe it would raise the quality and the readability of the paper.
Another question for the authors is if they thought about including a dichotomous key to the new species.

Experimental design

Experimental design
The paper is consistent with the Aims & Scope of the journal as it results from original research on the study area concerned (Brazil) and in the environmental sciences domain. The description of new species of nematodes is not only relevant to nematode taxonomists and ecologists, but also in general for the description of the overall benthic biodiversity.
The methods are clearly explained and valid, meeting the criteria of the journal. Nonetheless, a part of the Methods section has been omitted and the authors refer to a paper previously published on PeerJ for information on it (Manoel, Neres & Esteves 2024).
As I understand from that paper, the samples were obtained from 6 stations along the break of the continental shelf. How many replicates did you have for each of those stations? According to the rules of the journal, a meaningful replication is encouraged. Aside from the rules of the journal, unless the sampling is merely qualitative, a fair number of replicates should always be sampled.
Manoel, Neres & Esteves (2024) report that a box corer was used to collect the sediment, from which cores (10 cm x 10 cm) were obtained. Did you extract the meiofauna from all of that sediment or just an aliquot of it?
For the sake of the context, adding some basic information about the project and the campaign would be nice for the reader, as in the cited paper this kind of information is scant as well.
How many individuals of the new species did the authors find in their samples? If I’m not mistaken, you found 4 males and 2 females for Molgolaimus sigmoides sp.nov., 5 males and 2 females for Molgolaimus paralongispiculum sp.nov. and just 1 and 1 for Molgolaimus brevispiculum sp.nov. I assumed it from the data you included in your tables. It could be nice to have the information between brackets in the description of the species.
In the methods section (lines 100 and 101) the authors mention colloidal silica as the mean for the extraction of meiofauna. Can you be more specific? Was it Ludox? Which kind of (HS40 for instance)? How many times did you centrifuge the samples and for how long?

Validity of the findings

Validity of the findings
The description of the three new species is correctly supported by data on the holotype and several paratypes, both male and female. As far as I understand, for Molgolaimus brevispiculum sp.nov. the authors only had one female paratype aside from the holotype. It is the strict minimum for the description of a species, but this doesn’t make it any less valid.
The description of the species is supported by measurements, pictures (both bright field and SEM) and a section for unambiguous species determination, which make the paper robust enough for publication. The discussion provides critical elements concerning the division in subgroups of the species of Molgolaimus (Leduc, Fu & Zhao 2019) and proposes a different subdivision based on the amphideal fovea.

Additional comments

General comments
Line 35 : analyzes -> analyses
Line 59 : no need of « it » after the comma (“The order… it began”)
Line 95: Materials and methods section. In lines 89 and 90 you give some information about the sampling location and in line 96 you refer to the paper of Manoel, Neres & Esteves (2024) for the details. Nonetheless, I think the reader would appreciate having a bit more information about the study area, the habitat type and the sampling design. It’s not the core or the paper of course, so I wouldn’t indulge too much, but I’d organize the section as follows:
Study area and sampling. Country, location, habitat type, sediment type
Line 98: “Sample processing” probably sounds better than “Laboratory processing”
Line 186: Dezember  December
Line 182: I suggest “Type material” to name the paragraph
Line 185: I suggest “Type locality” to name of the paragraph
Line 204: Lightly or slightly, just pick one
Line 210: The use of “posterior” is inaccurate. You can say “Ventral gland located posterior to the pharyngo-intestinal junction” if behind is what you meant.
Line 211: To the left of the intestine
Line 223: To the right side of the intestine
Line 354: the intestine
Line 363: posterior to
Line 365: the… the
Line 378: Differential diagnosis. I would rephrase the paragraph for clarity as follows (feel free to modify some elements):
“Only males of Molgolaimus typicus Furstenberg & Vincx, 1992 and Molgolaimus drakus Fonseca, 381 Vanreusel & Decraemer, 2006 have been described, thus the comparison between those species and Molgolaimus brevispiculum sp. nov. is only based on males.
The absence of gubernaculum is a common characteristic between M. typicus, M. drakus and M. brevispiculum sp. nov., which are thus different from the other species of the genus Molgolaimus. Nonetheless, other features as the absence of precloacal supplements, the spicules length, the cephalic setae length, the size of the amphidial fovea in relation to the body diameter and the de Man’s ratio help in differentiating those three species (Table x).
In addition…”
I would then add a small table to summarize the differences between the species. Personally, I think it makes the paragraph and the section much more readable.
Line 406, 407: “thus making it impossible”… “thus making it impossible”. Please avoid the repetition.
Line 418: no comma needed between “spicules” and “is”
Line 423: “whose spicules length measure between 1 and 3 times the cloacal body diameter”
Line 458: “The remaining”… “remained” please avoid the repetition.
Line 470 to 472: Please rephrase the sentence, as it’s exactly the same as in the Handbook of Zoology
Lines 495 to 502: I found this last paragraph a bit repetitive. Here I would emphasize the strength and uniqueness of the paper, though in a more synthetic way. Your paper provides a description of three new species of Molgolaimus from the continental shelf break of Northeastern Brazil. You provide critical elements leading to a reorganization of the division in subgroups. Nonetheless, you had the minimum requirements for describing one of your species, thus a greater sampling effort in the region is needed and generally, more effort should be put in the positioning of the genus Molgolaimus in the classification.

---

## Round 0.2 · accepted · Accept

The authors have addressed all the comments and suggestions made by referee. The manuscript already in its first review stage was of high level; this was underlined by 4 reviewers who gave minor revisions. For this reason, I think that the paper is ready for publication